# The Network of Angiotensin Receptors in Breast Cancer

**DOI:** 10.3390/cells9061336

**Published:** 2020-05-27

**Authors:** Filippo Acconcia

**Affiliations:** Department of Sciences, Biomedical Sciences and Technology Section, University Roma TRE, Viale Guglielmo Marconi 446, I-00146 Rome, Italy; filippo.acconcia@uniroma3.it; Tel.: +39-0657336345; Fax: +39-0657336321

**Keywords:** angiotensin, breast cancer, AT1R, AT2R, MASR, MRGD

## Abstract

The renin-angiotensin system (RAS) is a network of proteins regulating many aspects of human physiology, including cardiovascular, pulmonary, and immune system physiology. The RAS is a complicated network of G-protein coupled receptors (GPCRs) (i.e., AT1R, AT2R, MASR, and MRGD) orchestrating the effects of several hormones (i.e., angiotensin II, angiotensin (1–7), and alamandine) produced by protease-based transmembrane receptors (ACE1 and ACE2). Two signaling axes have been identified in the RAS endocrine system that mediate the proliferative actions of angiotensin II (i.e., the AT1R-based pathway) or the anti-proliferative effects of RAS hormones (i.e., the AT2R-, MAS-, and MRGD-based pathways). Disruption of the balance between these two axes can cause different diseases (e.g., cardiovascular pathologies and the severe acute respiratory syndrome coronavirus 2- (SARS-CoV-2)-based COVID-19 disease). It is now accepted that all the components of the RAS endocrine system are expressed in cancer, including cancer of the breast. Breast cancer (BC) is a multifactorial pathology for which there is a continuous need to identify novel drugs. Here, I reviewed the possible roles of both axes of the RAS endocrine network as potential druggable pathways in BC. Remarkably, the analysis of the current knowledge of the different GPCRs of the RAS molecular system not only confirms that AT1R could be considered a drug target and that its inhibition by losartan and candesartan could be useful in the treatment of BC, but also identifies Mas-related GPCR member D (MRGD) as a druggable protein. Overall, the RAS of GPCRs offers multifaceted opportunities for the development of additional compounds for the treatment of BC.

## 1. Introduction

The renin-angiotensin system (RAS) plays a critical role in the regulation of human physiology. In fact, it is a master regulator of cardiovascular physiology that further integrates, coordinates, and allows the functioning of different physiological systems, including the renal and respiratory systems. Remarkably, RAS is a network of hormones and receptors, which generates a local endocrine system regulated by specific enzymes controlling and determining different and contrasting effects in different tissues and organs. Interestingly, as angiotensins are peptide hormones, most angiotensin receptors are G-protein coupled receptors (GPCRs) that cause the activation of several different signaling transduction pathways, allowing the effects of the hormones to show cell-type specificity [1]. Therefore, it is not surprising that the deregulation of this delicate endocrine balance could generate different pathologies, including atherosclerosis, ischemic disease, hypertension, and heart failure [2].

Classically, the protease renin acts on liver-produced angiotensinogen to generate angiotensin I (AngI), which is converted to AngII by angiotensin converting enzyme 1 (ACE1). Angiotensin II (AngII) can subsequently bind to two different GPCRs named AT1R and AT2R that mediate the physiological effects of this hormone. Interestingly, the activation of AT1R is related to the cardiovascular regulatory functions of AngII. Indeed, the resulting signaling mediates the vasoactive properties of AngII. However, excess AngII:AT1R signaling could lead to the above-mentioned pathological conditions [1,2,3].

The functions of the AngII-mediated activation of AT2R-based signaling counterbalance the effects of AngII:AT1R activation by fine-tuning the physiological effects of this hormone [4]. In recent years, it has become clear that an additional mechanism that counteracts AngII:AT1R signaling is due to the action of ACE2, which converts AngII into the hormone Ang(1–7), which is subsequently transformed into alamandine [5]. Ang(1–7) and alamandine in turn act on two additional GPCRs named the proto-oncogene Mas receptor (MASR) [6] and Mas-related GPCR member D (MRGD) [7]. Signaling triggered by the hormonal activation of these two receptors further buffers the potential detrimental effects of AngII:AT1R activation [8]. In turn, the balance between the relative activation of the AngII:AT1R axis and the AngII:AT2R/Ang(1–7):MASR/alamandine:MRGD axis is critical for cardiovascular and noncardiovascular physiology [8] (Figure 1). The importance of this counterbalancing axis emerged during the recent pandemic of severe acute respiratory syndrome coronavirus 2 (SARS-CoV-2), which has highlighted this concept. Indeed, SARS-CoV-2 uses the ACE2 receptor as the entry portal into host cells [9,10]. After being engaged by the spike protein of the virus, ACE2 is internalized and cannot convert any more AngII into Ang(1–7), which results in the consequent reduction in alamandine and leads to the accumulation of AngII. The consequent imbalance towards the activation of AngII:AT1R signaling leads to the systemic pathognomonic signs of coronavirus 19 disease (COVID-19) [9].

Interestingly, in addition to its well-known cardiovascular functions, the RAS system has been recognized to play a role in the pathogenesis and development of different tumors [11], including tumors of the breast [12]. Breast cancer (BC) is a heterogeneous disease with different molecular phenotypes, and a prominent role in breast carcinogenesis is played by 17β-estradiol (E2) signaling, which occurs through the engagement of estrogen receptor α (ERα). In fact, breast tumors are categorized as ERα-positive, which result in a better prognosis, or as ERα-negative, which usually result in a poorer prognosis [13,14]. ERα-positive tumors are treated with drugs aimed at interfering either with the availability of endogenous E2 (e.g., aromatase inhibitors) or with E2:ERα interactions (e.g., selective ER modulators (SERMs) such as 4OH-tamoxifen or selective ER modulators (SERDs)) [13]. However, from a systemic point of view, these compounds display severe side effects that, in the case of SERMs, can even induce tumor formation in other tissues or organs (i.e., endometrial cancer for 4OH-tamoxifen) and often generate resistant tumor cells that result in disease recurrence and tumor relapse [13]. Metastatic breast cancers (MBCs) remain diseases with unmet needs in terms of their pharmacological treatment. Therefore, research has been focused on developing strategies that could lead to the identification of novel drugs to treat primary BCs and MBCs [14].

In this respect, the RAS system has been recently proposed as a novel drug target in tumors [11] and in BC on the basis not only of its complicated molecular networks and pleiotropic functions in cells, but also in light of the fact that GPCRs are membrane receptors that can be targeted by different inhibitors or activators that could prove to be excellent tools to address the disease [1].

In this review, I provide an update on the possible functions of the components of the RAS endocrine system in BC by highlighting the possibility of using these components as target(s) for novel BC treatments.

## 2. AT1R and AT2R Signaling

The functions of AT1R and AT2R can be ascribed to their AngII-dependent differential activation of intracellular signaling in different cell contexts. Indeed, AngII-dependent activation of AT1R determines cell proliferation and cell hypertrophy and increases ROS production, while AngII:AT2R signaling is anti-proliferative and inhibits ROS production [5,15].

AngII:AT1R signaling has long been recognized, and consequently, the molecular pathways activated by hormones through AT1R have been described. In general, two different molecular arms can be identified for the molecular machinery induced by AngII binding (Figure 2). The binding of the hormone to AT1R triggers a classical GPCR-associated cascade as well as noncanonical signaling (for a detailed review, please see [5]). Activation of the AT1R receptor leads to the activation of the phospholipase (PL) C-, PLD-, and PLA2-based molecular pathways, which ultimately affect the activity of myosin-light chain kinase (MLCK), which is required for the cytoskeleton remodeling and consequently the cell contraction required for vasoconstriction. AngII also allows the activation of the classical ERK/MAPK pathways. Indeed, it has been shown that this hormone can induce cell proliferation, growth and survival as well as cardiac hypertrophy by increasing the activity of the ERK/MAPK and PI3K/AKT pathways through cross-talk with different receptor tyrosine kinases, such as the epidermal growth factor receptor. Activated AT1R also recruits nonreceptor tyrosine kinases such as Src that in turn allow the recruitment of a molecular complex containing Janus kinases (JAKs). In addition, following AngII administration, an increase in reactive oxygen species (ROS) has been demonstrated. ROS also contribute as signaling molecules to the regulation of the pathways required for both the induction of cell proliferation and the regulation of cytoskeletal contraction. Notably, the signaling network activated by the AngII:AT1R complex further regulates extracellular matrix production and favors cell adhesion [5]. It is worth noting here that not all the signaling pathways function in the same cellular context at the same time. Indeed, classical GPCR-based signaling occurs faster than noncanonical signaling, and the specificity of the activated signaling pathways depends on the cell type in which AngII acts [5,16] (Figure 2).

In contrast to AngII:AT1R signaling, the molecular signaling pathways elicited by the AngII:AT2R complex are less well studied and understood (Figure 2). The main signaling pathways activated by AT2R have been identified in the activation of protein phosphatases, the nitric oxide (NO)–cGMP system and PLA2/arachidonic acid signaling. The AngII:AT2R complex strongly activates three classes of protein phosphatases (MAP kinase phosphatase 1 (MKP-1), protein phosphatase 2A (PP2A), and SH2 domain-containing phosphatase (SHP-1)). These proteins determine the inhibition of the ERK/MAPK, PI3K/AKT, and JAK phosphorylation cascades. The stimulation of the NO–cGMP pathway shifts the balance of MLCK towards the inhibitory state and the activation of PLA2/arachidonic acid signaling, which leads to cross-talk with other signaling cascades and contributes to the activation of PP2A and the inactivation of ERK/MAPK. In turn, the activation of AngII:AT2R signaling is considered anti-proliferative and pro-apoptotic as well as being capable of inducing vasodilation [4].

Therefore, the physiological effects of AngII are finely modulated by the balance of the activity of AT1R and AT2R, which simultaneously influence intracellular signaling in a cell type-specific manner, allowing the cell specificity of the functions of AngII spatially and temporally (Figure 1).

## 3. Proto-Oncogene Mas Receptor (MASR) Activation and Mas-Related GPCR Member D (MRGD) Signaling

MASR and MRGD have been considered orphan GPCRs since their identification, and their cloning has been relatively recent (i.e., 1986 for MASR and 2002 for MRGD) [17,18]. Accordingly, the discovery of their association with the complexity of RAS signaling occurred in parallel with the discovery of two other AngII-derived peptides (i.e., Ang(1–7) and alamandine) [8] (Figure 1). Although there is still open debate regarding the possibility that Ang(1–7) and alamandine could be the only endogenous ligands of MASR and MRGD, respectively, it is generally believed that the activation of MASR by Ang(1–7) and the activation of MRGD by alamandine could contribute to the protective effects of these two peptides in the cardiovascular system [8]. In this respect, it is important to note that ACE2 action is the prerequisite for the formation of alamandine (Figure 1) since this hormone is derived from ACE2-produced Ang(1–7), which is the substrate for the aspartate decarboxylase (AD), the producer of alamandine [8].

Given the recent discovery of the receptors MASR and MRGD as well as their ligands, knowledge of the signaling pathways activated by the Ang(1–7):MASR and alamandine:MRGD complexes is still under investigation (Figure 3). However, Ang(1–7) stimulation of MASR determines the activation of signaling cascades that partially overlap with those elicited by the AngII:AT2R complex, thus contributing to the protective effects of this axis of the RAS endocrine system. Indeed, Ang(1–7) induces NO production through the activation of the cGMP-based pathway. Activation of diverse protein phosphatases leading to the inhibition of the ERK/MAPK pathway has also been reported as an action of Ang(1–7) (e.g., SHP-1). Additionally, Ang(1–7) can also elicit the activation of the PI3K/AKT signaling cascade [8].

Alamandine:MRGD signaling is even less well understood than Ang(1–7):MASR signaling (Figure 3). However, in the last five years, evidence has accumulated that indicates the alamandine-mediated activation of different signaling cascades. c-Src, p38/MAPK, and IκB activation and NO production have been observed [19]. The protein kinase (PK) C and PKA signaling pathways can be stimulated by the alamandine:MRGD complex [8,20]. Alamandine was also reported to induce the activation of the adenosine monophosphate kinase (AMPK) pathway [7], to inhibit the MAPK signaling pathways [21] and to activate Jun-*N*-terminal kinase (JNK) phosphorylation to inhibit the nuclear factor κB (NF-κB) signaling pathway [22]. Alamandine-dependent stimulation of NO via MRGD also determines the activation of calcium/calmodulin-dependent protein kinase II (CaMKII) [23]. Interestingly, the differential activation of the different pathways is cell type-specific and has been observed in adipose tissue and adipose tissue cells as well as in different cellular models of the cardiovascular apparatus [19].

Nonetheless, in all experimental settings, the effects of alamandine:MRGD complex activation elicit transduction cascades that mediate the almandine protective functions, which counteract the detrimental effects of AngII (Figure 1).

## 4. The RAS Endocrine System in Cancer

Given the complexity of the RAS signaling pathways, it should not be surprising that the deregulation of this delicate network involving cross-talk between 3 hormones and 4 receptors could play a role in the triggering and development of different diseases, including cancer. In this respect, the importance of the RAS endocrine system in different types of human cancers has highlighted the possibility of targeting these diseases by drugging the complex receptor network in the RAS pathways [1,11,12,24]. Remarkably, in BC, the components of the RAS system are expressed, thus suggesting both a role for this system in the development and progression of the disease [12] and a potential druggable pathway.

### 4.1. Breast Cancer

It is now accepted that BC is a disease with different molecular phenotypes. At diagnosis, BC is classified based on the presence of ERα, which is thought to be one of the main drivers of BC progression. Indeed, most BCs (70%) express this receptor, which mediates the proliferative effects of the hormone E2 [13]. ERα-expressing BCs often result in a more favorable prognosis than the remaining 30% of ERα-negative breast tumors, for which only limited pharmacological options are available [13]. BC cells often become “addicted” to E2 signaling via ERα, as it sustains proliferative signaling [25], favors cell survival and resistance to external stimuli-induced cell death [26,27], and supports cell motility, invasion, and metastasis [28,29].

Consequently, endocrine therapy (ET) is an elective choice for the treatment of ERα-expressing BCs. ET targets E2-dependent signaling through ERα either by drugs that reduce the availability of circulating E2 (e.g., aromatase inhibitors–AIs) or by inhibitors of ERα (i.e., 4OH-tamoxifen (Tam) and/or Faslodex (i.e., fulvestrant or ICI 182,780–FUL)) [13,30,31]. However, it is now clear that ET drugs have serious side effects that, despite the proven efficacy, undermine their use and result in the necessity to identify novel treatments for BCs. Indeed, although Tam extends the overall survival of approximately 85% of ERα-positive BC patients for as long as 5 years after diagnosis [32,33], prolonged Tam treatment causes in a large proportion of patients the development of cells resistant to this anti-ERα drug through different mechanisms [30,33,34,35,36,37,38,39,40,41]. In turn, resistant cells acquire a proliferative advantage that result in tumor recurrence and relapse of the pathology. Unfortunately, such metastatic Tam-resistant BCs are difficult diseases for which a standard treatment protocol still does not exist [42]. Consequently, important research efforts have been made in the search for new therapeutic approaches to identify novel drugs and/or drug targets with the aim of developing alternative strategies to treat ERα-expressing BC both in the context of primary disease and in the context of metastatic disease [14].

During the last 5 years, a great improvement in the understanding of the genes involved in the progression of BC was obtained from the frequent use of CRISPR/CAS9-based pooled loss-of-function genetic screens [43,44,45,46]. In these experiments, a particular cell line is simultaneously exposed to the reduction of the levels of all the mRNAs produced by the entire genome and then cultured for the desired period of time in the presence or the absence of compound treatment. At the end of the experimental window, through next-generation sequencing techniques, it is possible to identify those genes that are required for growth (i.e., Achille genes) or those that are not strictly important for cell survival (i.e., fitness genes) [45,46]. This approach complemented the development of multiple preclinical models of antiestrogen resistance that have been demonstrated to be invaluable tools to identify unexpected and potentially druggable novel pathways for the treatment of primary BCs and MBCs [33,44,47,48,49,50,51,52].

In the next few sections, an update of the research performed during the last 5 years regarding the impact of the components of the RAS system on BC is evaluated to understand the role of these components as novel potential drug targets for BC treatment.

#### 4.1.1. AT1R and Breast Cancer

Several connections have been made that link AT1R activity and expression to different aspects of BC pathogenesis, including carcinogenesis, development of the disease, and acquisition of the metastatic phenotype (Table 1).

Regarding carcinogenesis, AT1R polymorphisms have been discovered in a population of Indian women, and specific polymorphic genotypes have been demonstrated to contribute to a predisposition to develop more aggressive disease with an advanced stage and increased tumor size [53]. Furthermore, a bioinformatic analysis showed that the presence of AT1R in BC together with a set of 12 genes can be considered a marker to predict disease resulting in a poor prognosis [54]. Therefore, early identification of AT1R may represent an important diagnostic parameter that could help identify populations of women or patients at high risk to develop aggressive forms of BCs and in turn inform the selection of appropriate treatment plans for the specific patient [53,54].

Several reports have now clearly demonstrated that AT1R is overexpressed in a subset of breast tumors that express ERα but do not express the HER2 receptor and contributes to tumor development. More importantly, AT1R expression in these BC tumors is correlated with a poor outcome of the disease [55,56,57,58] and is a marker of chemotherapeutic resistance [1]. AT1R overexpression, which has oncogenic potential in the absence of AngII administration [1], has been documented in the luminal A and B subtypes of BC and correlated with aggressive characteristics, including increased lymph node metastasis (LMN), reduced responsiveness to neoadjuvant therapy (i.e., ET), and reduced overall survival. Stable overexpression of AT1R in MCF-7 cells confers growth advantages through the concomitant increased expression of diverse survival factors. The proliferative advantage parallels the increased motility and migration ability of MCF-7 cells. AT1R overexpression in MCF-7 cells also induces the activation of endothelial-to-mesenchymal transition (EMT) [58]. Remarkably, AT1R expression was also found to be increased in Tam-resistant MCF-7 cells that mimic metastatic BC disease [1]. In addition, it has been shown that the AT1R-dependent increase in BC LNM could depend on an increase in chemokine release that results in the activation of the focal adhesion kinase (FAK)/rhotekin A (RhoA) pathway and the consequent stimulation of cell migration and invasion and acquisition of the metastatic phenotype [56].

Given the close relationship between AT1R and BC, many investigators have demonstrated that AT1R antagonists (i.e., losartan and candesartan) can work as anti-BC drugs. Indeed, it has been shown that such compounds prevent all the oncogenic effects elicited by AT1R [55,56,57,58]. As noted above, there is an urgent need to identify novel BC drugs that could be used to treat both primary tumors and MBC, especially in light of the fact that it is imperative either to avoid the development of resistant phenotypes or to attempt to block Tam-resistant metastatic breast tumor growth. In this respect, different strategies are being used to identify novel BC treatments [14]. Our research group has recently introduced the concept of targeting the selective modulation of ERα levels as a promising approach to discover new anti-BC compounds [14,59,60,61]. We have indeed demonstrated that drugs that do not necessarily bind to ERα but instead change the amount of ERα protein in BC cells can prevent BC cell proliferation and subsequent tumor growth [14]. In turn, we demonstrated this by screening a library of 1018 Food and Drug Administration (FDA)-approved drugs in BC cells to search for molecules that modify ERα content and prevent BC cell growth [14,59,61,62]. This line of research led to the discovery that more than 50 FDA-approved compounds affected ERα content in MCF-7 cells and, among them, 11 also inhibited cell proliferation [59]. We further studied emetine, carfilzomib, and methotrexate, and showed that they block the proliferation of cellular models of primary BC and MBC more effectively than classic ET drugs (e.g., fulvestrant) [59,61,62]. Thus, this approach allowed us to identify molecules with “anti-estrogen-like” effects that we previously called EMERALDs (i.e., selective modulators of ERα levels and degradation) [14].

Surprisingly, careful examination of the drugs modulating ERα levels in BC cells further revealed that among the 50 drugs that changed receptor levels in MCF-7 cells, we found both losartan and candesartan [59]. Losartan and candesartan do not bind to ERα in vitro (unpublished data), and we did not observe a reduction in basal or E2-induced MCF-7 cell proliferation [59]. Although our observations seem to contradict the above-mentioned results, we treated MCF-7 cells with losartan and candesartan at nanomolar concentrations [59], while their effects were observed at micromolar concentrations [56]. In contrast, it is worth noting that the effect of the AT1R inhibitors on the ERα content in MCF-7 cells is reminiscent of that elicited by Tam, thus suggesting that these drugs could mimic the effects of this anti-estrogen on BC cells.

#### 4.1.2. AT2R and Breast Cancer

The role of AT2R in BC is not very well understood. This is indicated by the fact that a PubMed search using AT1R and BC as keywords retrieved 13 published papers in the last 5 years while using AT2R and BC to find literature data retrieved only 4 hits. In addition, contrasting results have been reported for the role of this GPCR in cancer in general and in BC in particular.

Indeed, data are available that connect AT2R and cell proliferation in BC. AT2R has been found to be overexpressed in breast disease as well as in Tam-resistant MCF-7 cells [1,63,64]. Interestingly, a bioinformatic analysis identified AT2R as a master regulator controlling diverse genes involved in several processes impacting known hallmarks of cancer [65]. AT2R has also been shown to be useful for gene therapy, as the AT2R gene has been delivered through a nasal spray in lung carcinoma patients and observed to be effective in attenuating cancer growth [66]. Accordingly, AT2R has shown proliferative effects that can be inhibited by specific antagonists in melanoma cells [67].

On the other hand, other reports have indicated that there is no relationship between AT2R and BC cell proliferation. Indeed, the proliferative effects of AngII are not mediated by AT2R in BC cell lines [68], and AT2R was not found to be associated either with the clinical-pathologic variables of BC or with the BC cell proliferation rate or angiogenesis [69].

Therefore, the evidence accumulated so far does not allow us to consider this GPCR as a critical player in BC development and progression (Table 1).

#### 4.1.3. MASR and MRGD in Breast Cancer

The roles of MASR and MRGD in BC have not yet been assessed. However, accumulating evidence suggests that the AngII:AT2R/Ang(1–7):MASR/alamandine:MRGD axis could function and therefore be potentially druggable in cancer [24].

Clues regarding the potential role of a given protein in cell proliferation can be extracted from the annotated information in freely accessible web databases, where the impact of the reduction of protein levels on cancer cell proliferation is revealed. One of these databases is DepMap (https://depmap.org/portal/), in which specific scores are assigned to genes that may or may not affect cell proliferation so that it is possible to identify those genes required or dispensable for growth (i.e., Achille and fitness genes, respectively) [45,46]. Inspection of the DepMap database indicates that MASR and MRGD are not Achille genes (i.e., reduction in their protein levels does not impact cell proliferation) for most cell lines. Interestingly, while MASR is not an Achille gene for cancer cells (Table 1), MRGD is important for cell proliferation in cell lines derived from hematopoietic and lymphatic tissue, cells of the upper GI tract, and cells of the skin. According to this observation, MRGD is tumorigenic and is highly expressed in lung cancer [6].

A similar approach can be undertaken by studying the annotated data obtained from BC cells under different experimental conditions [43,44]. Recently, the work published by the group of Dr. Carroll has provided an experimental framework to identify genes important for the development of Tam resistance in BC cells [44]. Surprisingly, although MASR is not an Achille gene for BC cells according to the DepMap database, in the presence of Tam, the lack of MRGD induces a strong anti-proliferative response in MCF-7 cells [44] (Table 1). Thus, it is tempting to speculate that MRGD signaling could support the activation of the survival pathways required for the initial steps leading to resistance to this antiestrogen.

## 5. Discussion and Conclusions

The knowledge accumulated so far regarding the RAS endocrine system clearly indicates that this regulatory network is composed of two different axes that regulate opposing effects that impact the balance between the induction of cell proliferation and the activation of the anti-proliferative program. Similar to other hormonal systems (e.g., E2- and ER-based signaling) [70], AngII exerts its pleiotropic effects by acting through AT1R and AT2R to activate opposing signaling pathways (Figure 1 and Figure 2). However, in parallel, cells have evolved a two-receptor-based system (i.e., MASR and MRGD) activated by two additional hormones (e.g., Ang(1–7) and alamandine) that are derived from the metabolization of AngII through the activity of ACE2 and AD, respectively. Remarkably, the molecular events activated by this second axis affect the same signaling pathways activated by the AngII:AT2R complex and reinforce the anti-proliferative effects of the RAS system (Figure 1 and Figure 3).

Disruption of the equilibrium between the AngII:AT1R and the AngII:AT2R/Ang(1–7):MASR/alamandine:MRGD axes can lead to an imbalance in the control of cellular and human physiology, as witnessed by the critical effects induced in humans infected by SARS-CoV-2 [9]. Not surprisingly, therefore, at least some of the components of the RAS endocrine system play a role in some aspects of tumor progression and development. Interestingly, in BC, AT1R appears to be involved in the development of the disease during tumorigenesis as well the development of the metastatic phenotype, while to date, no strong evidence has been found to definitively link AT2R to such pathology (Table 1).

In turn, AT1R has already been considered a drug target in different types of tumors, including those of the breast [16]. In fact, the anti-neoplastic activity of losartan and candesartan has been evaluated. Interestingly, although different mechanisms have been reported for the action of these AT1R inhibitors [16], our work also suggests that losartan and candesartan could act as selective modulators of ERα levels and degradation (i.e., EMERALDs) [14,59] to prevent BC cell proliferation. Future investigations aimed at verifying this hypothesis are required. However, it is tempting to speculate that losartan and candesartan could mimic the effect of ET drugs (e.g., stabilizing ERα and/or inhibiting it, as in the case of Tam), thus functioning as “anti-estrogen-like” compounds. In this way, these drugs could be added to the repertoire of compounds used to fight BC.

The AT2R-based pathway appears not to confer a selective advantage in tumor cells. Remarkably, this phenomenon could easily be explained by the fact that AngII-based signaling through AT2R is anti-proliferative; thus, hyperactivation of the protective branch of the RAS system would prevent tumor cell proliferation. Accordingly, MASR does not confer a growth advantage to tumor cells. However, the evaluation of the requirement of MRGD for tumor cell proliferation through the available annotated data (e.g., DepMap and [44]) indicates that this receptor could be involved not only in tumor growth, but also in the process required for the development of resistance to chemotherapy, as in the case of Tam in BC cells. Clearly, although this observation requires further investigation and validation, it also presents the possibility that the AngII:AT2R/Ang(1–7):MASR/alamandine:MRGD axis could be considered as a drug target [24], at least in BC. In this respect, it is worth pointing out that there is great interest in drugs that, rather than inhibiting a pro-survival or proliferative pathway, can act by hyperactivating an anti-proliferative signaling cascade; the net result of such an activator would be superimposable on the action of a specific inhibitor. Accordingly, the protective branch of the RAS network, whose components are expressed in cancers [12], could be considered a Trojan horse for tumor cells.

In conclusion, the data available so far in the literature regarding the interactions among the components of the RAS endocrine system and BC suggest that the receptors of both axes can be used as drug targets in a differential manner, especially in light of the fact that most of them are GPCRs for which activators or inhibitors are either available or relatively simple to rationally design. Nonetheless, additional studies are required to understand the molecular functions of these pathways to manipulate them for novel cancer treatment.

## Figures and Tables

**Figure 1 cells-09-01336-f001:**
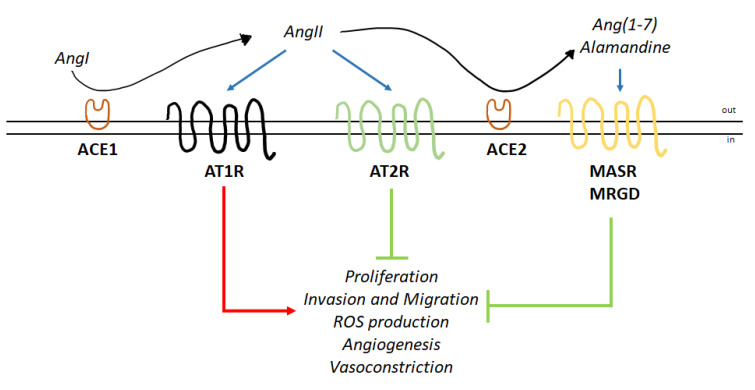
Angiotensin II induces effects through AT1R and AT2R. Schematic of the functions elicited by the two axes of the renin-angiotensin system (RAS) endocrine network. Angiotensin II (AngII) produced by ACE1-dependent action on angiotensin I (AngI) binds to AT1R and AT2R. AngII is transformed by ACE2 into Ang(1–7) and subsequently into alamandine. Ang(1–7) and alamandine bind to the proto-oncogene Mas receptor (MASR) and Mas-related GPCR member D (MRGD), respectively. The blue arrows indicate binding to receptors; the red arrow indicates activation; and the green arrows indicate inhibition (for details, please see the text).

**Figure 2 cells-09-01336-f002:**
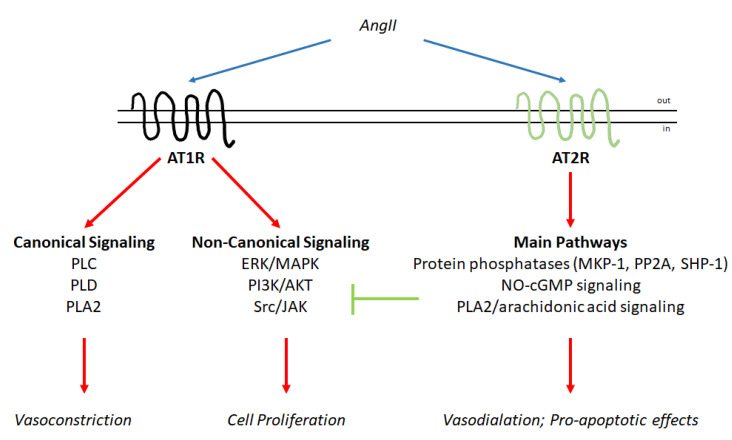
AT1R and AT2R signaling networks. Schematic of the signaling cascades activated by the binding of angiotensin II (AngII) to AT1R and AT2R. The blue arrows indicate binding to receptors; the red arrow indicates activation; and the green arrow indicates inhibition (for details, please see the text).

**Figure 3 cells-09-01336-f003:**
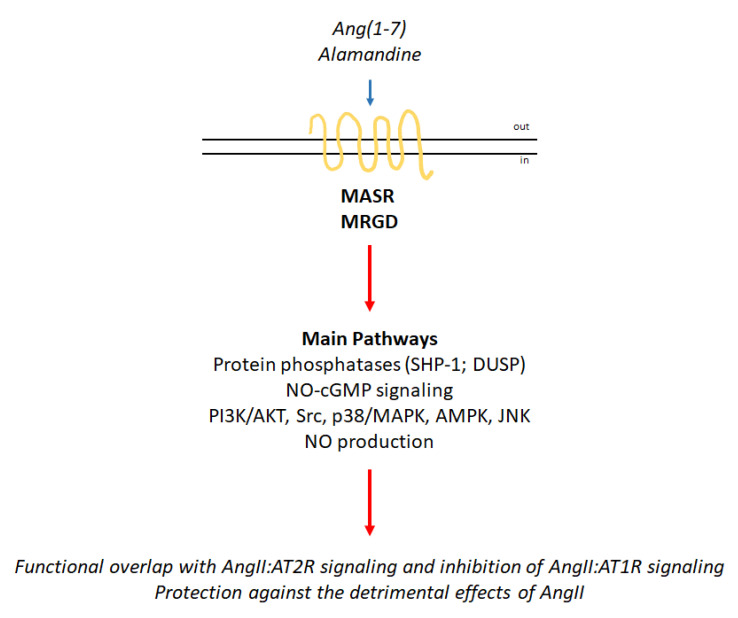
MASR and MRGD signaling network. Schematic of the signaling cascades activated by the binding of angiotensin(1–7) (Ang(1–7)) to MASR and MRGD. The blue arrow indicates binding to receptors; the red arrows indicate activation (for details, please see the text).

**Table 1 cells-09-01336-t001:** Roles of the components of the RAS system in breast cancer. For details and references, please see the text.

Components of the RAS System	Role in Breast Cancer
AT1R	Linked to carcinogenesis, development of the disease and acquisition of the metastatic phenotype.
AT2R	Evidence for contrasting connections.
MASR	None yet identified.
MRGD	Putatively connected with acquisition of resistance to endocrine therapy.

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
