# Peer review of "The Network of Angiotensin Receptors in Breast Cancer"

_cells, 2020, doi:10.3390/cells9061336_

Round 1

Reviewer 1 Report

I would like to suggest the author to include some schematic diagrams as well as tables in this manuscript. The author needs to pay significant attention to English writing to improve the manuscript. In addition, some latest research papers on the subject seem missing. 

Author Response

Reviewer #1

I would like to suggest the author to include some schematic diagrams as well as tables in this manuscript.

Author Response: I thank this Reviewer for this comment. I have now added 3 figures and 1 table.

The author needs to pay significant attention to English writing to improve the manuscript.

Author Response: The manuscript has been corrected by American Journal Expert for professional English editing.

In addition, some latest research papers on the subject seem missing.

Author Response: We updated the reference section.

Reviewer 2 Report

In this review the author provide an update on the functions of the components of the Renin Angiotensin System in breast cancer. He also suggests to use these components as a pharmacological target(s) in breast cancers.

The topic is of interest and the text is well-written. In my opinion, the manuscript gives real contribution to research that is worth to be published in Cells

Minor revisions:

The author should insert figures illustrating the cellular signalling described in the review

Author Response

Reviewer #2

In this review the author provide an update on the functions of the components of the Renin Angiotensin System in breast cancer. He also suggests to use these components as a pharmacological target(s) in breast cancers.

The topic is of interest and the text is well-written. In my opinion, the manuscript gives real contribution to research that is worth to be published in Cells.

Author Response: I thank this Reviewer for this comment.

Minor revisions:

The author should insert figures illustrating the cellular signalling described in the review

Author Response: I thank this Reviewer for this comment. I have now added 3 figures and 1 table.

Reviewer 3 Report

The manuscript is difficult to read and to follow the idea of it since it seems disorganized (eg it refers to the COVI infection, but later the topic is not discussed). Also, a revision of the English language is necessary.
References 1 and 2 seem to be interchanged. A general review of references will be appropriated. An example is reference 3, a review article. In the present manuscript, the statements derived from the review article are not reflected with their correct reference. On page 5, only two references are shown on line 96.
The authors describe that "AngII subsequently can bind to two different GPCRs named AGTR1 and AGTR2 that mediates the physiological effects of this hormone" however, AGTR1 and 2 refer to the genes encoding AT1R and AT2R (Renziehausen et al 2019).
Finally, note that the renin-angiotensin system is a set of effector peptides regulated by specific enzymes. Therefore, the consideration of "RAS is now considered as a network of hormones" must be specifically referenced.
The use of abbreviations also requires revision. Absence of section 4.1.1.
An explanatory scheme/drawing of the different routes and interactions that are intended to be exposed will be necessary.

Author Response

Reviewer #3

The manuscript is difficult to read and to follow the idea of it since it seems disorganized (eg it refers to the COVI infection, but later the topic is not discussed).

Author Response: The aim of the present review was to provide an update of the last 5 years on the functions of the components of the renin angiotensin system in breast cancer. To this purpose, I first summarized the actual knowledge of the RAS signaling and then analyzed if the components of the RAS signaling is present and druggable in BC cells. In particular, as outlined in section 4.1.2, I performed a PubMed search using all the components of the RAS system and BC as keywords. I found (and wrote) that a Pubmed search using AT1R and BC as keyword retrieved 13 published papers in the last 5 years while using AT2R and BC to find literature data retrieved only 4 hits. Moreover, even less hits are present regarding the relationship between MASR and MRGD in breast cancer, thus I analyzed published dataset to hypothesize their potential role in BC, which I discussed in the discussion section. Nonetheless, at the end of the section 4.1, I wrote that “In the next few sections, an update of the research performed during the last 5 years regarding the impact of the components of the RAS system on BC is evaluated to understand the role of these components as novel potential drug targets for BC treatment.” in order to clarify the aim of the subsequent sections.

Regarding, SARS-COV2 infection, there is not any need to discuss in detail this point as it will be out of the scope of the review. The reference to this virus and to the COVID-19 disease was done to stress the importance of the RAS system. Nonetheless, I also recalled viral disease in the discussion section: “Deregulation of the equilibrium between the AngII:AT1R and the AngII:AT2R/Ang(1-7):MASR/alamandine:MRGD axes can determine an imbalance of the control of cellular and human physiology, as witnessed by the critical effects of the SARS-Cov2 infection”.

Also, a revision of the English language is necessary.

Author Response: The manuscript has been corrected by American Journal Expert for professional English editing.

References 1 and 2 seem to be interchanged.

Author Response: I thank the Reviewer and changed the references accordingly.

A general review of references will be appropriated. An example is reference 3, a review article. In the present manuscript, the statements derived from the review article are not reflected with their correct reference. On page 5, only two references are shown on line 96.

Author Response: I have now amended and updated the reference section as per this Reviewer request.

The authors describe that "AngII subsequently can bind to two different GPCRs named AGTR1 and AGTR2 that mediates the physiological effects of this hormone" however, AGTR1 and 2 refer to the genes encoding AT1R and AT2R (Renziehausen et al 2019).

Author Response: I thank the Reviewer for raising this point. I amended the test by changing AGTR1 with AT1R and AGTR2 with AT2R.

Finally, note that the renin-angiotensin system is a set of effector peptides regulated by specific enzymes. Therefore, the consideration of "RAS is now considered as a network of hormones" must be specifically referenced.

Author Response: I agree that I did not specify the fact that the RAS is regulated by specific enzymes in this sentence. However, effector peptides that bind to specific receptors and activates intracellular signalling to physiological responses are, by definition, hormones. In my opinion, this concept does not require any specific reference as it is textbook knowledge. The amended sentence is now: “RAS is a network of hormones and receptors, which generates a local endocrine system regulated by specific enzymes…”.

The use of abbreviations also requires revision.

Author Response: I have clarified the acronym for which I did not made explicit the abbreviation.

Absence of section 4.1.1.

Author Response: I have corrected this mistake and re-formatted the bullet points accordingly.

An explanatory scheme/drawing of the different routes and interactions that are intended to be exposed will be necessary.

Author Response: I thank this reviewer for this comment. I have now added 3 figures and 1 table.

Round 2

Reviewer 1 Report

The manuscript has been improved.